# Analysis of Prior Aspirin Treatment on in-Hospital Outcome of Geriatric COVID-19 Infected Patients

**DOI:** 10.3390/medicina58111649

**Published:** 2022-11-15

**Authors:** Khaoula Zekri-Nechar, José Barberán, José J. Zamorano-León, María Durbán, Alcira Andrés-Castillo, Carlos Navarro-Cuellar, Antonio López-Farré, Ana López-de-Andrés, Rodrigo Jiménez-García, Carlos H. Martínez-Martínez

**Affiliations:** 1Medicine Department, School of Medicine, Universidad Complutense de Madrid, 28040 Madrid, Spain; 2Internal Medicine Department HM Hospital, 28250 Madrid, Spain; 3Public Health and Maternal and Child Health Department, School of Medicine, Universidad Complutense de Madrid, IdISSC, 28040 Madrid, Spain; 4Statistics Department, Universidad Carlos III, 28903 Madrid, Spain; 5Maxillofacial Surgery Department, Hospital General Universitario Gregorio Marañón, 28007 Madrid, Spain

**Keywords:** aspirin, COVID-19, elderly population, low molecular weight heparin, pulmonary thromboembolism, mortality, hospital stay

## Abstract

*Background and Objectives*: Aspirin (ASA) is a commonly used antithrombotic drug that has been demonstrated to reduce venous thromboembolism. The aim was to analyze if geriatric COVID-19 patients undergoing a 100 mg/day Aspirin (ASA) treatment prior to hospitalization differ in hospital outcome compared to patients without previous ASA therapy. *Materials and Methods*: An observational retrospective study was carried out using an anonymized database including geriatric COVID-19 patients (March to April 2020) admitted to Madrid Hospitals Group. A group of COVID-19 patients were treated with low ASA (100 mg/day) prior to COVID-19 infection. *Results*: Geriatric ASA-treated patients were older (mean age over 70 years; *n* = 41), had higher frequency of hypertension and hyperlipidemia, and upon admission had higher D-dimer levels than non-ASA-treated patients (mean age over 73 years; *n* = 160). However, patients under ASA treatment did not show more frequent pulmonary thromboembolism (PE) than non-ASA-treated patients. ASA-treated geriatric COVID-19-infected patients in-hospital < 30 days all-cause mortality was more frequent than in non-ASA-treated COVID-19 patients. In ASA-treated COVID-19-infected geriatric patients, anticoagulant therapy with low molecular weight heparin (LMWH) significantly reduced need of ICU care, but tended to increase in-hospital < 30 days all-cause mortality. *Conclusions*: Prior treatment with a low dose of ASA in COVID-19-infected geriatric patients increased frequency of in-hospital < 30 days all-cause mortality, although it seemed to not increase PE frequency despite D-dimer levels upon admission being higher than in non-ASA users. In ASA-treated geriatric COVID-19-infected patients, addition of LMWH therapy reduced frequency of ICU care, but tended to increase in-hospital < 30 days all-cause mortality.

## 1. Introduction

Thrombotic complications and a hypercoagulable state are frequent in SAR-CoV-2 (COVID-19), strongly contributing to mortality. Venous thrombotic embolisms (VTE) have been observed in patients who were otherwise asymptomatic and in hospitalized COVID-19-infected patients in whom up 20% to 30% of critically ill patients could develop VTE [1]. Moreover, published retrospective series of cases of COVID-19 patients have shown frequent elevation of D-dimer, which has been related to acute pulmonary thrombosis, dramatically worsened the prognosis [2].

Several reports have shown that older people were at a higher risk of COVID-19 complications, with higher rates of hospitalization, intensive care unit admissions, and death [3,4]. Among other proposed hypotheses to explain the higher vulnerability of older people to worse COVID-19 prognosis and higher frequency of hospitalization there have been a weaker immune response, obesity, fragility, etc. [5]. At present, it is evident that age of people infected by COVID-19 has decreased, probably because a greater number of older people have already received their vaccination. However, at present, it is difficult to predict the future ability of vaccines to prevent possible new mutations in the virus that may confer them immunity escape and increased infectivity. Therefore, it is important to not stop analyzing effects and impact of currently used antithrombotic drugs.

Aspirin (ASA) is without doubt the main antithrombotic drug in the world to prevent thrombotic events including venous thrombosis. For example, the multinational and prospective Pulmonary Embolism Prevention (PEP) study, where patients undergo surgery for hip fracture or elective arthroplasty, found that 160 mg of ASA daily reduced the risk of symptomatic VTE by ~36% when compared with placebo [6]. Other studies as Warfarin and Aspirin (WARFASA) and the Aspirin to Prevent Recurrent Venous Thromboembolism (ASPIRE) trials, in which patients underwent initial anticoagulation therapy before randomly being assigned to low-dose ASA or placebo, also demonstrated significant reduction in VTE recurrence with ASA [7,8].

ASA is often prescribed for secondary prevention in patients with cardiovascular diseases and other comorbidities that, in older patients with COVID-19, might be associated with higher mortality [9]. However, to our knowledge, there are very few studies analyzing the effect of ASA on the outcome of COVID-19-infected patients, and even less in the older population. Moreover, in most of the studies, ASA was added as in-hospital active treatment against COVID-19. In this regard, for example, a work carried out in critically ill COVID-19 patients analyzed the effects of in-hospital treatment of ASA combined with low molecular weight heparin (LMWH) [10]. Paradoxically, this study concluded higher incidence of VTE and worse in-hospital outcome after combined treatment of ASA with a therapeutic dose of LMWH [10]. Another study also suggested that there was no significant difference in terms of mortality under ASA treatment of COVID-19-infected patients [11]. The study did not clearly report whether the use of ASA was an active prescription or a history of exposure [11]. Moreover, the possible influence of ASA in the older population infected by COVID-19 was not previously analyzed.

Therefore, the main aim of the present study is to retrospectively analyze if geriatric COVID-19 patients taking daily low ASA (100 mg/day) doses prior to hospitalization have their <30 days in-hospital outcome affected compared to non-ASA-treated geriatric patients.

## 2. Materials and Methods

### 2.1. Included Patients

This retrospective study was carried out using an anonymized database provided by Madrid Hospitals Group. Data are from COVID-19-infected patients admitted to their hospital net who kindly made available their medical information to some Spanish researchers within the program: COVID-19 data saves lives in Spain.

The study included 201 consecutive patients aged 60 admitted for hospitalization between 7 March and 4 April, 2020 (initiation of the first COVID-19 wave in Spain) with clinical and radiological data compatible with COVID-19 (On 25 March the Spanish Health Secretary recommended not to confirm COVID-19 infection by reverse transcriptase polymerase chain reaction (RT-PCR) when clinical and radiological presentation was typical). Therefore, only in patients included before these data or patients without the typical clinical and radiological presentation COVID-19 infection was confirmed by RT-PCR.

Exclusion criteria were hospital stays >30 days and admission platelet count lower than 25 × 109 platelets/L. These criteria were based on guidelines recommendation of prophylactic dose of LMWH in COVID-19 patients and the follow-up of patients with short hospitalization stay [12,13].

As clinical outcomes, parameters considered included <30 days in-hospital all-cause mortality, requirement of admission to intensive care unit (ICU), diagnosis of pulmonary embolism (PE), length of hospital stay 30 days longer. PE was assessed by pulmonary computed tomography (CPTA). Upon admission, C-reactive protein and D-dimer levels were also considered.

COVID-19-infected patients admitted for hospitalization were considered and divided into two groups: 1. COVID-19-infected patients who were taking cardiovascular doses of ASA (100 mg/day) prior to COVID-19 infection and 2. COVID-19-infected patients without prior ASA treatment. Only the patients who were taking ASA before hospital admission continued receiving 100 mg/day ASA during hospitalization.

From admission and during hospitalization some patients received LMWH (bemiparin sodium, Rovi Lab. Madrid. Spain. 2.500–10.000 UI). Patients who, upon admission, had not been treated with LMWH did not receive anticoagulating therapy during their hospitalization course.

The study was approved by the local Ethical Committee (Code: 21/084-E. Approval date: 17 February 2021) and conducted in accordance with the Declaration of Helsinki.

### 2.2. Statistical Analysis

Categorical variables were expressed as frequency and percentage and compared by Chi-square test. Continuous variables were expressed as mean ± standard error of mean (S.E.M). Student t-test was used to compare quantitative variables. A *p* value < 0.05 was considered significant. The statistical analysis was performed with the SPSS software version 25.0.

## 3. Results

### 3.1. Comparison among Patients under ASA Treatment before Hospital Admission

The mean age of the studied COVID-19 patients was over 73 years old (Table 1). COVID-19 patients under ASA treatment before hospital admission were slightly but statistically significantly older than the patients who were not undergoing ASA treatment prior hospitalization (Table 1). Gender was not different among analyzed geriatric COVID-19 patients that were or were not taking ASA (Table 1).

Upon admission, geriatric COVID-19 patients under ASA treatment more frequently had hypertension or dyslipidemia compared to geriatric patients without previous ASA therapy (Table 1). There was a similar frequency of diabetic mellitus patients between the two analyzed groups.

In the in-hospital follow up, patients taking ASA prior to hospitalization had a higher frequency of <30 days all-cause mortality than patients without ASA treatment (Table 1). The other analyzed parameters such as need of ICU care, PE development, and length of hospital stay were not different among the geriatric COVID-19-infected patients that were or were not taking ASA before hospitalization (Table 1).

Upon admission, D-dimer levels, but not C-reactive protein, were significantly higher in COVID-19 patients taking ASA than in those without ASA (Table 1). It was noteworthy that in patients with prior ASA treatment, upon-admission D dimer levels were similar among patients discharged alive and those who died in <30 days of follow up (D-dimer upon admission µg/L: ASA-treated dead patients: 2.18 ± 0.63; ASA-treated discharged patients: 1.97 ± 0.48, *p* = 0.494). Moreover, in the group of ASA-treated COVID-19 patients, in-hospital < 30 days all-cause mortality was more frequently observed in patients showing higher C-reactive protein levels upon admission (C-reactive protein upon admission (mg/L): ASA-treated < 30 days dead patients: 139.02 ± 26.12; discharged patients: 83.73 ± 16.11, *p* = 0.033).

### 3.2. Effect of LMWH Therapy in Geriatric Patients Who Were Taking ASA before Hospitalization

A number of geriatric patients (*n* = 32) taking ASA before hospitalization were treated with LMWH upon admission (Table 2). As Table 2 shows, these LMWH-treated geriatric ASA patients were of similar age and gender compared to those ASA patients who were not treated with LMWH (Table 2). It should be pointed out that the patients who, upon admission, were not treated with LMWH never received LMWH during hospitalization follow up. Moreover, four patients taking ASA were from admission treated with therapeutic doses of LMWH (5.000–10.000 IU) continuing this dosage during the follow-up. In addition, 28 of the 32 analyzed ASA-treated patients, upon admission received prophylactic dosage of LMWH (2.500–3.500 IU) but during in-hospital follow up in 6 of them LMWH dosage was increased reaching therapeutic LMWH dose.

In addition, with the aim of identifying the possible additional effect of anticoagulant treatment, closely associated with antiplatelet treatment with ASA, on <30 days in-hospital outcome, it was analyzed potential effect of LMWH treatment on hospital admission in geriatric COVID-19-infected patients taken daily Aspirin (100 mg/day) (Table 2). Geriatric COVID-19 patients taking ASA who received LMWH therapy less frequency required ICU care than patients taking ASA who did not receive LMWH (Table 2). Length hospital stay was similar between the geriatric COVID-19 patients taking ASA treated or no with LMWH. Frequency of PE tended to be slightly higher in the patients taking ASA that were treated with LMWH although it did not reach statistical significance (Table 2). Among geriatric COVID-19 ASA patients, <30 days all-cause in-hospital mortality almost reached statistics significant as comparing those who were treated or not with LMWH. In fact, ASA patients who did not receive LMWH tended to show less frequency of <30 days all-cause mortality that patients taking ASA who received LMWH therapy (Table 2).

Upon admission D-dimer and C-reactive protein levels were similar among patients taking ASA who were treated or not with LMWH (Table 2).

Two geriatric COVID-19 patients taking ASA undergoing to LMWH treatment had a hemorrhagic event (drop in hemoglobin levels > 5 g/dL but without clinical hemorrhage sign).

## 4. Discussion

This retrospective study suggested that geriatric patients who were taking a low daily ASA dose (100 mg/day) prior to COVID-19 infection and during hospitalization showed higher frequency of <30 days all-cause mortality than those geriatric COVID-19 patients not taking ASA before or during the hospitalization course. As compared with COVID-19-infected geriatric patients not previously treated with ASA, patients under ASA treatment showed higher D-dimer levels upon admission. However, the incidence frequency of PE was not different among them.

The World Health Organization reported that over 95 % of fatalities caused by COVID-19 in Europe had been individuals aged over 60 [14]. In Spain, during the most critical period of the first wave of COVID-19 disease, older age was the main risk factor for severity of COVID-19 disease [15]. Although mean age of infected patients in the following COVID-19 waves was significantly lower, probably because a larger proportion of older people were vaccinated, it is not possible to discard that COVID-19 mutations may favor the infection of these particular special vulnerable population. Therefore, while waiting for new specific drugs against COVID-19 to be developed, it is important to continue to study more “classic” commonly used drugs that may have a certain effect against the deleterious effects of COVID-19 infection.

It is recognized that viral infections, as COVID-19 infections, are commonly accompanied by platelet activation and aggregation [16]. In this regard, it was reported that ASA had an impact on both DNA and RNA viruses, although, certainly, these studies were carried out in vitro and using high ASA doses [17,18].

The elderly population has a high frequency of use of ASA within their daily treatments. However, to our knowledge, in the elderly population, possible associations between prior use of low-dose ASA and in-hospital progression of COVID-19 infection were not analyzed. In fact, the reported studies about effects of ASA on COVID-19-infected patients were carried out in critically ill COVID-19 patients of a wide range of ages, suggesting that patients under ASA treatment had higher VTE incidence and mortality [19]. However, a meta-analysis of six eligible studies which included patients of all ages concluded that use of low-dose ASA was independent of reduced mortality of COVID-19 patients [20].

In the present study, geriatric COVID-19 patients taking ASA (100 mg/day) prior to hospital admission, with the daily ASA treatment continuing during hospitalization follow up, had higher <30 days in-hospital all-causes mortality than those patients not taking ASA. Contrarily, different studies reported that active prescription of low-dose aspirin during or prior to hospitalization was associated with reduced risk of mortality among patients with COVID-19 [20,21]. These apparently paradoxical results could be related to difference in age of population and/or difference in the prevalence of cardiovascular comorbidities, which have been widely associated with worse COVID-19 outcome [22,23]. Following the same line of evidence, in COVID-19 patients with coronary artery disease no differences in <30 days all-cause mortality was found among those taking low dose ASA and those without ASA treatment [24].

COVID-19 disease has been associated with increased risk of VTE [25]. COVID-19-infected geriatric patients under ASA treatment showed higher D-dimer levels upon admission than those without ASA treatment. However, PE frequency was similar among them. It could be related to evidence showing low dose ASA may reduce VTE [26].

Guidelines recommend a prophylactic dose of LMWH as treatment for all COVID-19 patients requiring hospitalization, in the absence of any contraindications such as active bleeding and platelet count lower than 25 × 109 platelets/L [12]. Interestingly, a recent report has identified heparan sulfate, a glycosaminoglycan molecule like heparin, as coreceptor for COVID-19, further supporting that exogenous heparin may provide therapeutic benefits for patients with COVID-19 infection [27]. Moreover, as mentioned above, the WARFASA and ASPIRE trials demonstrated that, in patients under anticoagulant therapy, low-dose ASA led to a significant reduction in VTE [7,8]. Therefore, we analyzed if early addition of LMWH to ASA-treated geriatric COVID-19-infected patients may improve their hospital outcome.

In geriatric patients taking ASA, anticoagulant treatment was significantly associated only with reduced frequency of ICU care as compared with ASA-treated patients who did not receive LMWH. In LMWH-treated patients taking ASA, >30 days all-cause mortality tended to be higher than in those ASA patients who did not receive LMWH therapy, although it did not reach statistical significance probably due to the limited sample size. In this regard, there are many discrepancies in the results of COVID-19 patients treated with anticoagulation with respect to promotion of changes in mortality risk. Previous studies which included all ages of COVID-19-infected patients have reported reduction of in-hospital mortality as a result of anticoagulant therapy, particularly in patients with severe disease [28]. However, other reports have found no differences in mortality between heparin users and nonusers, and there was an observed mortality reduction in critically ill patients [29], although, even in these critical COVID-19 populations treated with prophylatic or therapeutic dose LMWH, controversial results were reported [30,31]. In addition, it was analyzed if the effect of LMWH treatment on the mortality may be based on the condition of patients (ICU care or not ICU care). Results revealed that mortality significantly decreased in patients who took LMWH without ICU care (76% vs. 24%); however, this reduction was not observed in those patients under LMWH who needed ICU care (48% vs. 52%). This interesting finding was supported by previous studies [32,33].

In this regard, authors suggest that D-dimer levels should guide more aggressive thromboprophylaxis regimens using higher doses of heparin, although studies have also reported that the mortality rate of patients from both prophylactic and therapeutic LMWH-treated groups was similar when patients were classified according to D-dimer results [31]. In our study, D-dimer levels upon admission were similar among geriatric patients taking ASA prior to COVID-19 infection who were treated or not with LMW during hospitalization.

### Study Limitations and Comments

The present study has some limitations and the results should be interpreted with caution. As mentioned, data was obtained from an anonymized database made freely available to some researchers by the Madrid Hospital Group and, therefore, recruitment of new data to increase sample size was not possible. Therefore, sample size is a limiting factor for the result interpretation. In addition, confounding factors may exist that were not included in the database. In this regard, patients subjected to aspirin treatment usually have, as a main indication, the involvement of peripheral, coronary, neurologic, or renal arteriosclerosis. Therefore, it would be plausible to find higher mortality in those patients who were not taking aspirin prior to hospital admission. In addition, several factors influencing death in patients with COVID-19 were not considered. However, it would be plausible to consider the admissions to ICU as approximation of severity of COVID-19 and, therefore, of factors influencing death in patients with COVID-19. Relative to the patients taking ASA, a study limitation is that we did not know how long they were under the antithrombotic treatment prior to hospitalization for COVID-19 and their compliance to the therapeutic regimen. However, all these patients were taking daily ASA before being infected with COVID-19. Moreover, we could not assess if the platelets of all the included ASA-treated patients responded to ASA. In this regard, it is demonstrated that patients with ASA-resistant platelets have higher incidence of thrombotic events than patients with ASA-sensitive platelets [34]. Being aware of these limitations, there is no doubt about the need to carry out specific studies in one of the populations most vulnerable to COVID-19, the elderly population.

## 5. Conclusions

To conclude, geriatric COVID-19 patients who underwent prior antiplatelet treatment with ASA had a higher frequency of in-hospital < 30 days all-cause mortality than non-ASA-treated patients. It could be probably related to higher frequency of comorbidities associated with worse prognosis of COVID-19. Moreover, despite ASA-treated patients having higher D-dimer levels upon admission, they showed similar frequency of PE compared to COVID-19-infected geriatric patients who were not taking ASA, despite the fact that the latter had lower D-dimer levels upon hospital admission. Administration of LMWH to geriatric COVID-19-infected patients taking ASA before hospitalization reduced the frequency of ICU care, but tended to increase < 30 days all-cause mortality with respect to patients taking ASA and who, during hospitalization, never received LMWH therapy.

## Figures and Tables

**Table 1 medicina-58-01649-t001:** Effect of previous Aspirin treatment on in-hospital outcome of geriatric COVID-19-infected patients.

Variables			Previous Aspirin Treatment		
			Not *n* = 160 (%)	Yes *n* = 41 (%)	*p* Value
Age (years)			73.54 ± 0.63	76.05 ± 1.11	0.041
Gender	Men		99 (61.9)	26 (63.4)	0.856
	Women		61 (38.1)	15 (36.6)	
Diseases	Hypertension	No	96 (60.0)	17 (41.5)	0.033
		Yes	64 (40.0)	24 (58.5)	
	Hyperlipidaemia	No	105 (65.6)	17 (41.5)	<0.001
		Yes	55 (34.4)	24 (58.5)	
	Diabetes mellitus	No	144 (90.0)	34 (82.9)	0.204
		Yes	16 (10.0)	7 (17.1)	
ICU	No		113 (70.6)	27 (65.9)	0.553
	Yes		47 (29.4)	14 (34.1)	
Length of Hospital Stays (days)			11.64 ± 0.53	11.05 ± 1.02	0.623
PE	No		119 (74.4)	32 (78.0)	0.627
	Yes		41 (25.6)	9 (22.0)	
Mortality	Survivals		113 (70.6)	27 (65.9)	0.014
	Death		47 (29.4)	14 (34.1)	
D-Dimer (µg/mL)			1.66 ± 0.24	2.02 ± 0.37	0.011
C-Reactive Protein (mg/L)			111.68 ± 8.48	106.46 ± 14.28	0.871

Age, length of hospital stays, D-dimer, and C-reactive protein values are expressed as mean ± SE.

**Table 2 medicina-58-01649-t002:** Effect of LMWH treatment on hospital admission in geriatric COVID-19-infected patients taken daily Aspirin (100 mg/day).

Variables			LMWH Administration on Hospital Admission		
			Not *n* = 9 (%)	Yes *n* = 32 (%)	*p* Value
Age (years)			74.11 ± 2.413	76.59 ± 1.25	0.360
Gender	Men		3 (33.3)	12 (37.5)	0.819
	Women		6 (66.7)	20 (62.5)	
Diseases	Hypertension	No	6 (66.7)	11 (34.4)	0.082
		Yes	3 (33.3)	21 (65.6)	
	Hyperlipidaemia	No	4 (44.4)	13 (40.6)	0.837
		Yes	5 (55.6)	19 (59.4)	
	Diabetes mellitus	No	8 (88.9)	26 (81.3)	0.591
		Yes	1 (11.1)	6 (18.8)	
ICU	No		1 (11.1)	26 (81.3)	<0.001
	Yes		8 (88.9)	6 (18.7)	
Length Hospital Stays (days)			9.4 ± 1.4	11.50 ± 1.25	0.670
PE	No		9 (100)	23 (71.9)	0.072
	Yes		0 (0)	9 (28.1)	
Mortality	Survivals		8 (88.9)	17 (53.1)	0.052
	Death		1 (11.1)	15 (46.9)	
D-Dimer (µg/mL)			1.41 ± 0.25	1.94 ± 0.40	0.565
C Reactive protein (mg/L)			68.42 ± 15.73	114.93 ± 17.80	0.312

Age, Length hospital stays, D-dimer and C-reactive protein values are expressed as mean ± SEM. may have a footer.

## Data Availability

This study was carried out using an anonymized database provided by Madrid Hospitals Group. Data are from COVID-19-infected patients admitted in their hospital net that they made kindly available to some Spanish researchers within the program: COVID-19 data saves lives in Spain.

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
