# Peer review of "Analysis of Prior Aspirin Treatment on in-Hospital Outcome of Geriatric COVID-19 Infected Patients"

_medicina, 2022, doi:10.3390/medicina58111649_

Round 1
Reviewer 1 Report
The study is important, but patients who used aspirin usually have as main indication the involvement of peripheral, coronary, neurologic or renal arteriosclerosis. Therefore, there are diseases associated with the main causes of cardiac mortality or stroke. Mortality may be higher in this group due to these associations. The association with heparin-based anticoagulation is another unquestionable benefit in covid 19. I suggest increasing the number of patients if possible in future studies
Author Response
COMMENT
The study is important, but patients who used aspirin usually have as main indication the involvement of peripheral, coronary, neurologic or renal arteriosclerosis. Therefore, there are diseases associated with the main causes of cardiac mortality or stroke. Mortality may be higher in this group due to these associations. The association with heparin-based anticoagulation is another unquestionable benefit in covid 19. I suggest increasing the number of patients if possible in future studies
ANSWER
Thank you very much for the constructive revision that will surely improve the manuscript.
We absolutely agree with reviewer´s observation, patients subjected to aspirin treatment usually have diseases which are associated with higher mortality risk respect to group without aspirin. According to reviewer´s interesting observation, it has been considered as limitation of our results, being added the following sentence “….In this regard, patients subjected to aspirin treatment usually have as main indication the involvement of peripheral, coronary, neurologic or renal arteriosclerosis. Therefore, it would be plausible to find higher mortality respect to those patients who were not taking aspirin prior to hospital admission. In addition, several factors influencing death in patients with COVID-19 were not recruited. However, it would be plausible to consider the admissions to ICU as approximation of severity of COVID-19 and, therefore, factors influencing death in patients with Covid-19….” as limitation of our results in the discussion section of the revised manuscript.
In future studies, we will try to increase the sample size with the aim of obtaining results with higher statistical powerful.

Reviewer 2 Report
I thank the Editors for the possibility to review this paper and I thank the Authors for their work.
Aspirin is a commonly used antiplatelet drug. However direct oral anticoagulant was gradually used as the first-line medication to reduce venous thromboembolism in recent years. In the present study, the authors reported an observational study to analyze the outcome of prior aspirin treatment in old adult Covid-19 infected patients.
The topic of the work is interesting, but some important points need to be fixed to make it more complete.
Major compulsory revision.
The flaw of this study is that too many confounding factors are missing. To analyze the mortality for hospitalized patients, the parameters about why the patients use aspirin, or the other factors cause the death of patients with COVID-19 infection were all missing. The authors only report hypertension, hyperlipidemia, and diabetes mellitus. The other reasons to use aspirin, such as stroke, cardiovascular disease, atrial fibrillation, and deep venous thrombosis are missing. Besides, the factors influencing death in patients with COVID-19 are not reported too. ( Such as vaccination or not, oxygen use, steroid treatment or not. )
Minor revision
In the method part, the author showed "Exclusion criteria were hospital stays >30 days and on admission platelet count lower than 25x109 platelets/L" Why those patients were excluded from the study. The authors should clarify.
The authors analyze the effect of LMWH treatment on hospital admission in 41 Covid-19-infected patients taken daily Aspirin (100 mg/day). Why the author analyzed only 41 patients rather than use total 201 patients. The authors should clarify.
The author concluded that "Administration of LMWH to old adult Covid-19 infected patients taking ASA reduced the frequency of ICU care but tended to increase <30 days all-cause mortality with respect to patients taking ASA who during hospitalization never received LMWH therapy." Are the patients taking ASA during admission or did the patient have "Prior" aspirin treatment? The authors should clarify.
Author Response
Thank you very much for the detailed and constructive revision that will surely improve the manuscript.
COMMENT
Major compulsory revision.
The flaw of this study is that too many confounding factors are missing. To analyze the mortality for hospitalized patients, the parameters about why the patients use aspirin, or the other factors cause the death of patients with COVID-19 infection were all missing. The authors only report hypertension, hyperlipidemia, and diabetes mellitus. The other reasons to use aspirin, such as stroke, cardiovascular disease, atrial fibrillation, and deep venous thrombosis are missing. Besides, the factors influencing death in patients with COVID-19 are not reported too. ( Such as vaccination or not, oxygen use, steroid treatment or not.)
ANSWER
We thank the reviewer´s indications. Unfortunately, it was not able to recruit different factors potentially associated with the death of patients with Covid-19 infections. At current, it is not posible to obtain factors influencing death in Covid-19 patients due to anonymous aspect of the database. In addition, we absolutely agree with reviewer´s observations, patients subjected to aspirin treatment usually have diseases which are associated with higher mortality risk respect to group without aspirin. However, it would be plausible to consider the variable “admissions to the intensive care unit (ICU)” as approximation of severity of Covid-19 and, therefore, of factors influencing death in patients with Covid-19.
Therefore, it has been added as limitation of our results in the discussion section of the revised manuscript.:
“….In this regard, patients subjected to aspirin treatment usually have as main indication the involvement of peripheral, coronary, neurologic or renal arteriosclerosis. Therefore, it would be plausible to find higher mortality respect to those patients who were not taking aspirin prior to hospital admission. In addition, several factors influencing death in patients with Covid-19 were not recruited. However, it would be plausible to consider the admissions to ICU as approximation of severity of Covid-19 and, therefore, factors influencing death in patients with Covid-19 ….”
COMMENT
Minor revision
In the method part, the author showed "Exclusion criteria were hospital stays >30 days and on admission platelet count lower than 25x109 platelets/L" Why those patients were excluded from the study. The authors should clarify.
ANSWER
We thank reviewer´s indications. The hospitalization stay lower than 30 days was choosen based on previous studies, being considered as long hospitalization stay higher than 30 days (Signes-Costa J, et al. HOPE COVID-19 investigators. Prevalence and 30-Day Mortality in Hospitalized Patients With Covid-19 and Prior Lung Diseases. Arch Bronconeumol. 2021;57:13-20). The number of platelet was choosen according to guidelines recommendation of prophylactic dose of LMWH as treatment of all Covid-19 patients requiring hospitalization, in the absence of any contraindications such as active bleeding and platelet count less than 25x109 platelets/L (Thachil J, Tang N, Gando S, Falanga A, Cattaneo M, Levi M, et al. ISTH interim guidance on recognition and management of coagulopathy in COVID-19. J Thromb Haemost. 2020;18:1023-6).
With the aim of improving the understanding of readers, the following sentence was added in the Materials and Methods section of the revised manuscript: “…. Exclusion criteria were hospital stays >30 days and on admission platelet count lower than 25x109 platelets/L. These criteria were based on guidelines recommendation of prophylactic dose of LMWH in Covid-19 patients and the follow-up of patients with short hospitalization stay (12,13)…”
COMMENT
The authors analyze the effect of LMWH treatment on hospital admission in 41 Covid-19-infected patients taken daily Aspirin (100 mg/day). Why the author analyzed only 41 patients rather than use total 201 patients. The authors should clarify.
ANSWER
We thank reviewer´s observation. In the prsente study, the aim was to analyze the effect of ASA treatment on hospital outcomes in old adult Covid-19 patients. The reason of focusing the analysis on patients subjected to Aspirin treatment was based on that anticoagulant treatment (closely associated with antiplatelet treatment) may provide additional beneficial effects to antiplatelet treatment with ASA. We understand that effect of LMWH independently of ASA may have been analyzed, and it is an excellent idea. However, it was not the aim of the present study. In order to clarify this aspect, the following sentence has been added in the result section (3.2 subsection) of the revised manuscript:
“…In addition, with the aim of identifying the possible additional effect of anticoagulant treatment, closely associated with antiplatelet treatment with ASA, on <30 days in-hospital outcome, it was analyzed potential effect of LMWH treatment on hospital admission in old adult Covid-19-infected patients taken daily Aspirin (100 mg/day) (Table 2)…”.
COMMENT
The author concluded that "Administration of LMWH to old adult Covid-19 infected patients taking ASA reduced the frequency of ICU care but tended to increase <30 days all-cause mortality with respect to patients taking ASA who during hospitalization never received LMWH therapy." Are the patients taking ASA during admission or did the patient have "Prior" aspirin treatment? The authors should clarify.
ANSWER
We would like to apologyze for the lack of preccision in the sentence referred by the reviewer. This conclusion was referred to those patients who were taking aspirin before hospitalization. Therefore, the sentence “Administration of LMWH to old adult Covid-19 infected patients taking ASA reduced the frequency of ICU care but tended to increase <30 days all-cause mortality with respect to patients taking ASA who during hospitalization never received LMWH therapy” has been mofified by “Administration of LMWH to old adult Covid-19 infected patients taking ASA before hospitalization reduced the frequency of ICU care but tended to increase <30 days all-cause mortality with respect to patients taking ASA who during hospitalization never received LMWH therapy” of the new revised manuscript.

Reviewer 3 Report
This manuscript reports the analysis of prior aspirin treatment on in-hospital outcome of old adult covid-19 infected patients. This is an interesting study and the information could be useful to the related research and clinical application.
Comments:
1. Compared with literature in the similar study, the patient number in this study is too low and the conclusion is contradict with literature. Wijaya et al., reported the effects of aspirin on the outcome of COVID-19 with a total of 34,415 patients the systematic review and meta-analysis (Clin Epidemiol Glob Health. 2021, 12: 100883. doi: 10.1016/j.cegh.2021.100883). Martha et al. reported the active prescription of low-dose aspirin during or prior to hospitalization and mortality in COVID-19 using the data from 13,993 patients (Int J Infect Dis. 2021 Jul; 108: 6–12. doi: 10.1016/j.ijid.2021.05.016). Both studies concluded the use of aspirin was significantly associated with a reduced risk of mortality among patients with COVID-19, which is contradict with the conclusion from your study.
2. Regarding the application of LMWH in the treatment of COVID-19, two clinical trials (critically or noncritically ill) were reported: “In critically ill patients (based on 1098 patients) with Covid-19, an initial strategy of therapeutic-dose anticoagulation with heparin did not result in a greater probability of survival to hospital discharge or a greater number of days free of cardiovascular or respiratory organ support than did usual-care pharmacologic thromboprophylaxis”. (N Engl J Med 2021; 385:777-789, DOI: 10.1056/NEJMoa2103417); In noncritically ill patients (based on 2219 patients) with Covid-19, an initial strategy of therapeutic-dose anticoagulation with heparin increased the probability of survival to hospital discharge with reduced use of cardiovascular or respiratory organ support as compared with usual-care thromboprophylaxis (N Engl J Med 2021; 385:790-802, DOI: 10.1056/NEJMoa2105911). It seems the effect of LMWH treatment is based on the condition of patients (critically or noncritically ill). No patient condition (critically or noncritically) was provided in your study.
Author Response
Thank you very much for the detailed and constructive revision that will surely improve the manuscript.
COMMENT
This manuscript reports the analysis of prior aspirin treatment on in-hospital outcome of old adult covid-19 infected patients. This is an interesting study and the information could be useful to the related research and clinical application. 1. Compared with literature in the similar study, the patient number in this study is too low and the conclusion is contradict with literature. Wijaya et al., reported the effects of aspirin on the outcome of COVID-19 with a total of 34,415 patients the systematic review and meta-analysis (Clin Epidemiol Glob Health. 2021, 12: 100883. doi: 10.1016/j.cegh.2021.100883). Martha et al. reported the active prescription of low-dose aspirin during or prior to hospitalization and mortality in COVID-19 using the data from 13,993 patients (Int J Infect Dis. 2021 Jul; 108: 6–12. doi: 10.1016/j.ijid.2021.05.016). Both studies concluded the use of aspirin was significantly associated with a reduced risk of mortality among patients with COVID-19, which is contradict with the conclusion from your study.
ANSWER
We thank to reviewer for providing us interesting findings, since these studies will provide higher quality to the discussion section of the manuscript. In this regard, both studies have been cited and discussed in the new version of the manuscript. As you can see, it has been added the following paragraph in the discussion section: “….Contrarily, different studies reported that active prescription of low-dose aspirin dur-ing or prior to hospitalization was associated with reduced risk of mortality among patients with COVID-19 (20,21) This apparently paradoxical results could be related to difference age of population and/or difference in the prevalence of cardiovascular comorbidities, which have been widely associated with worse Covid-19 outcome (22,23)….”
COMMENT
2. Regarding the application of LMWH in the treatment of COVID-19, two clinical trials (critically or noncritically ill) were reported: “In critically ill patients (based on 1098 patients) with Covid-19, an initial strategy of therapeutic-dose anticoagulation with heparin did not result in a greater probability of survival to hospital discharge or a greater number of days free of cardiovascular or respiratory organ support than did usual-care pharmacologic thromboprophylaxis”. (N Engl J Med 2021; 385:777-789, DOI: 10.1056/NEJMoa2103417); In noncritically ill patients (based on 2219 patients) with Covid-19, an initial strategy of therapeutic-dose anticoagulation with heparin increased the probability of survival to hospital discharge with reduced use of cardiovascular or respiratory organ support as compared with usual-care thromboprophylaxis (N Engl J Med 2021; 385:790-802, DOI: 10.1056/NEJMoa2105911). It seems the effect of LMWH treatment is based on the condition of patients (critically or noncritically ill). No patient condition (critically or noncritically) was provided in your study. It has been added in the new method section of the revised manuscript. In addition,
ANSWER
We again thank to reviewer for providing us interesting studies. Unfortunately, in our study it is not possible to establish the variable “critical or noncritical ill”. However, according to reviewer´s interesting consideration, it would be plausible to consider the variable “admission to the intensive care unit (ICU)” as approximation of severity of Covid-19 (critically or noncritically ill). In this regard, it was performed an additional analysis to compare mortality in patients subjected to LMWH treatment who needed or not admissions to the ICU. This additional analysis revealed that, as reviewer suggested, the effect of LMWH treatment was based on the condition of patients. It has been added the following paragraph in the discussion section of the revised version of the manuscript: “….In addition, it was analyzed if the effect of LMWH treatment on the mortality may be based on the condition of patients (ICU care or not ICU care). Results revealed that mortality significantly decreased in patients who took LMWH without ICU care (76% vs 24%), however this reduction was not observed in those patients under LMWH who needed ICU care (48% vs 52%). This interesting finding was supported by previous studies (31,32)…..

Round 2
Reviewer 2 Report
The authors have responded to all my questions. I have no further comments. However, to understand whether prior aspirin treatment influences the outcome of old Covid-19 infected patients, prospective research may be needed.
Reviewer 3 Report
The revised MS has addressed my comments/concerns and it is OK to be accepted.